# NPF and NRT2 from *Pisum sativum* Potentially Involved in Nodule Functioning: Lessons from *Medicago truncatula* and *Lotus japonicus*

**DOI:** 10.3390/plants13020322

**Published:** 2024-01-22

**Authors:** Marie-Christine Morère-Le Paven, Thibault Clochard, Anis M. Limami

**Affiliations:** Univ Angers, Institut Agro, INRAE, IRHS, SFR 4207 QuaSaV, 49000 Angers, France; thibault.clochard@univ-angers.fr (T.C.); anis.limami@univ-angers.fr (A.M.L.)

**Keywords:** *Lotus japonicus*, *Medicago truncatula*, nitrate transporter, nodules, NPF, NRT2, *Pisum sativum*

## Abstract

In addition to absorbing nitrogen from the soil, legumes have the ability to use atmospheric N_2_ through symbiotic nitrogen fixation. Therefore, legumes have developed mechanisms regulating nodulation in response to the amount of nitrate in the soil; in the presence of high nitrate concentrations, nodulation is inhibited, while low nitrate concentrations stimulate nodulation and nitrogen fixation. This allows the legumes to switch from soil nitrogen acquisition to symbiotic nitrogen fixation. Recently, particular interest has been given to the nitrate transporters, such as Nitrate Transporter1/Peptide transporter Family (NPF) and Nitrate Transporter 2 (NRT2), having a role in the functioning of nodules. Nitrate transporters of the two model plants, *Lotus japonicus* and *Medicago truncatula*, shown to have a positive and/or a negative role in nodule functioning depending on nitrate concentration, are presented in this article. In particular, the following transporters were thoroughly studied: (i) members of NPF transporters family, such as LjNPF8.6 and LjNPF3.1 in *L. japonicus* and MtNPF1.7 and MtNPF7.6 in *M. truncatula*, and (ii) members of NRT2 transporters family, such as LjNRT2.4 and LjNRT2.1 in *L. japonicus* and MtNRT2.1 in *M. truncatula*. Also, by exploiting available genomic and transcriptomic data in the literature, we have identified the complete PsNPF family in *Pisum sativum* (69 sequences previously described and 21 new that we have annotated) and putative nitrate transporters candidate for playing a role in nodule functioning in *P. sativum*.

## 1. Introduction

Legumes are commonly used in sustainable agroecosystems because of their ability to tolerate low N fertilizer input due to their capacity to use atmospheric N_2_ through biological nitrogen fixation (BNF). The advantage of using legumes in agroecosystems is not limited to protecting soils from pollution caused by chemical fertilizers [1] because once well-established legumes progressively fertilize the soil [2]. Legumes, such as Pea (*Pisum sativum*), are nowadays introduced in cropping systems to provide ecological services i.e., limiting the usage of N fertilizer and decreasing herbicide input by competing with weeds for soil water, mineral nutrients and light, thus limiting their development [3,4].

Competitive genotypes to fulfil this role should be selected on the basis of their ability to efficiently colonize the soil with deep-foraging, fast-growing and highly branched root systems. These traits are known to be under the control of rhizosphere factors, among which nitrate as a signal molecule, sensed by various nitrate transporters such as NPF (Nitrate Transporter1/Peptide transporter Family) and NRT2 (Nitrate Transporter 2), plays a major role [5,6,7,8]. Paradoxically, if nitrate is necessary to ensure legumes’ seedling establishment before BNF starts, it is also a negative regulator of nodulation and BNF if it is provided at high concentrations [9]. For these reasons, increasing our knowledge of molecular aspects pertaining to nitrate sensing via nitrate transporters and signaling in legumes is a cornerstone for selecting genetically competitive genotypes suitable for ecological intercropping systems. 

Recently, particular interest has been given to the role of nitrate transporters in the functioning of nodules, with some transporters having a positive and/or a negative role in nodule functioning depending on nitrate concentration. This article updates the results obtained in the two model legumes, *Lotus japonicus* and *Medicago truncatula*. In addition, we have identified the complete PsNPF family in Pea using *P. sativum* v1a genomic assembly [10]. Thus, we were able to find 90 putative PsNPF sequences, among which we not only found the 69 previously described in the literature [11] but also identified 21 new sequences that we have annotated according to the two-number code [12]. Furthermore, we have also exploited available transcriptomic data in the literature generated in this species [13] to identify transporters, belonging to either NPF or NRT2 families, expressed in nodules that would be involved in positive or negative regulation in relation to nitrate concentration. 

## 2. Nitrogen Acquisition by Legumes

Most of the nitrogen taken up by higher plants is in inorganic form with nitrate as the major source. In their natural habitat, plants are exposed to frequent changes in mineral nutrient availability. In particular, to respond to the variations of nitrate availability in the soil, plants’ absorption mechanism of nitrate has evolved into two transport systems, the low-affinity transport system (LATS) and the high-affinity transport system (HATS) [14]. LATS proteins are mainly represented by NPF and HATS proteins are mainlyrepresented by NRT2 [7]. NPF members belong to a large family of 92 MtNPF in *M. truncatula* and 86 LjNPF in *L. japonicus* [15,16,17]. A study using the genomic data of 31 plant species, including *M. truncatula*, showed that NPFs belonged to eight subfamilies, a distribution which was confirmed for the NPFs of *L. japonicus* [12,17]. Using heterologous expression system, often Xenopus oocytes, some NPFs have been shown to be nitrate transporters, but others are likely to transport substrates like peptides, amino acids, glucosinolates, IAA or ABA, for example. Some NPFs were shown to be able to transport two different substrates [12]. NRT2s belong to a smaller family than NPF; in *M. truncatula*, this family includes three members [18]: MtNRT2.1 (Medtr4g057890), MtNRT2.2 (Medtr4g057865) and MtNRT2.3 (Medtr8g069775). In *L. japonicus,* this family consists of four members [19,20]: LjNRT2.1 (Lj3g3v3069030), LjNRT2.2 (Lj3g3v3069050), LjNRT2.3 (Lj4g3v1085060) and LjNRT2.4 (Lj1g3v3646440). However, LjNRT2.2 was shown to be not functional in some *L. japonicus* ecotypes because a stop codon interrupts the reading phase and results in a truncated protein [21]. Thus, it is reasonable to consider that NRT2 family of *L. japonicus* consists of three functional genes. All NRT2-type transporters transport only nitrate; the transport of this substrate requires, in most cases, the interaction of NRT2 with another protein, NAR2. Two NAR2 genes were identified in *M. truncatula*. While only a single NAR2 gene was identified in *L. japonicus* [8,22].

In addition to absorbing nitrate from the soil, legumes can form symbioses with bacteria, called rhizobia. The formation of root nodules allows legumes to perform atmospheric nitrogen (N_2_) fixation. In root nodule cells, rhizobacteria are enclosed in symbiosomes, which are structures surrounded by a peribacteroidal membrane (PBM) of plant origin. Bacteria differentiated into bacteroids acquire the ability to fix atmospheric N_2_ through nitrogenase enzymatic activity. Nitrogen fixation is a process requiring carbon energy supplied by the plant in the form of photosynthesis products, as well as oxygen for respiration to generate ATP and reducing power for the reduction of N_2_ to NH_3_. Paradoxically, if mitochondria require a normal level of O_2_ (normoxic condition) for respiration, then nitrogenase is inactivated by oxygen. This potential problem is solved thanks to the presence of leghemoglobin (Lb). This oxygen-carrying protein plays an important role; due to its high affinity for oxygen, it efficiently delivers oxygen to mitochondria of the bacteroids, while by buffering free oxygen, it decreases its level in the vicinity of nitrogenase [23]. Furthermore, Lbs protect nitrogenase as a scavenger of nitric oxide (NO), which is an inhibitor of its activity [24]. Proteins of the plant play a role in the infection and organogenesis, among which the NODULE INCEPTION (NIN) is one of the most important nodulation proteins. NINs are transcription factors that positively regulate rhizobial infection, nodule organogenesis and N fixation [25,26]. NINs also control nodule number by inducing expression of CLAVATA3/ENDOSPERM SURROUNDING REGION (CLE) peptides involved in communication between the root and shoot [27]. 

Symbiotic nitrogen fixation and nodule formation are energetically costly for the plant. Therefore, legumes have developed mechanisms regulating nodulation in response to the amount of nitrate in the soil [9]; in the presence of high nitrate concentrations, nodulation is inhibited. The responsiveness to high nitrate concentrations (5–10 mM) of nodule functioning has been associated with a decrease in functional leghemoglobin and nitrogenase activity [28]. In *M. truncatula* and *L. japonicus*, it has also been shown that NIN-LIKE PROTEIN (NLP) transcription factors play a central role in inhibiting nodulation under high nitrate [29,30,31,32]. On the contrary, low nitrate concentrations stimulate nodulation and nitrogen fixation. C-TERMINALLY ENCODED PEPTIDEs (CEPs) are signaling molecules that enhance nodulation [33,34]. In *M. truncatula*, MtCEP1 is induced under low nitrogen and expresses during nodule formation [33]. MtCEP1 has been shown to interact with its putative CRA2 (COMPACT ROOT ARCHITECTURE 2) receptor to mediate nodulation [34]. Those examples of mechanisms regulating nodulation in response to the amount of nitrate in the soil allow the legumes to switch from soil nitrogen acquisition to symbiotic nitrogen fixation.

While the inhibitory effect of nitrate at high concentration has often been studied on the formation, development and functioning of nodules, few studies have been dedicated to the positive effect of nitrate at low concentration. Omics studies have shown that the expression of many *NPF* genes is upregulated in mature nodules [35,36,37]. Some nitrate transporters of *L. japonicus* or *M. truncatula* have been shown to play a role in the functioning of nodules; some transporters have a positive and/or a negative role in nodule functioning depending on nitrate concentration. 

## 3. NPFs Playing a Role in Nodule Functioning

In *L. japonicus*, an in silico analysis showed that the expression of eight *LjNPF* genes was upregulated in mature N_2_-fixing nodules [36]. Two of these eight NPFs, LjNPF8.6 and LjNPF3.1, were studied in depth [38,39]. *LjNPF8.6*, whose expression is strongly induced in nodules compared to roots, is the first NPF for which a specific and positive role on nodule functioning has been shown [38]. *LjNPF8.6* was found to be located in the central infection zone where N fixation takes place [35]. In addition, after inoculation of *Ljnpf8.6* mutants by *Mesorhizobium loti*, an increase in nodular superoxide content in the nodules accompanied by a reduction in N-fixation activity was observed with an accumulation of anthocyanin in stems and roots [38]. Anthocyanin accumulation in stems has been reported as a phenotype associated with nitrogen starvation condition associated with impaired nodule function or lack of nodulation ([39] and references therein). These observations suggest that LjNPF8.6 plays a role in the control of nodule functioning rather than in development. Furthermore, this transporter was shown to have a nitrate transport activity; it is thus tempting to suggest that LjNPF8.6 plays a role in the control of nodule functioning through the modulation of nitrate flux trough the peribacteroidal membrane [38]. Another interesting transporter in *L. japonicus* is LjNPF3.1 [36]. The *LjNPF3* promoter was shown to be active in the cortical cells of inoculated hairy roots and at the base of the nodules [39]. Actually, its expression was more than 10-fold higher in nodules than in roots, while it was also expressed in leaves and mature flowers. In addition, inoculated *Ljnpf3.1* mutants showed increased nodule biomass and anthocyanin accumulation in the stems, phenotypes that can be explained by a slight but significant decrease in the measured nitrogenase activity. Thus, LjNPF3.1 plays a positive role in efficient nodule functioning, possibly by transporting nitrate from the roots or from outside to the nodules [39]. However, the role of LjNPF3.1 would be limited to conditions of low external nitrate concentration that are not inhibitory for BNF. 

In *M. truncatula*, the expression of several *MtNPFs* is upregulated in nodules [15]. However only two NPFs playing a role in nodule functioning, MtNPF1.7 and MtNPF7.6, have been deeply studied in *M. truncatula*. MtNPF1.7 (also known as LATD/NIP) was functionally characterized as a high-affinity nitrate transporter [40], involved in root development [41,42], with an essential role in the formation and maintenance of nodule meristems and in rhizobial invasion [42]. Studies of different mutants, affected in *MtNPF1.7*, have shown that MtNPF1.7 is not necessary for the initial stages of rhizobial invasion into host roots but is required for rhizobial infection during nodulation [43,44,45]. Since *MtNPF1.7* is expressed and required in both lateral root and nodule meristems, the corresponding protein could play a key role in the balance between development of lateral roots and nodules [42].

MtNPF7.6 is a NPF of *M. truncatula* studied in detail, specifically expressed in nodule vasculature, localized in the plasma membrane of nodule transfer cells (NTCs) [46]. Using knockout *mtnpf7.6* mutants, it has been shown that MtNPF7.6 modulates *Lb* expression, endogenous NO homeostasis and nitrogenase activity. MtNPF7.6 has been shown to play a role in nitrate-mediated regulation during root nodule symbiosis under both low- and high-nitrate conditions [46]. Under low nitrate (0.2 mM), MtNPF7.6, demonstrated as being a high-affinity nitrate transporter, functions in nitrate uptake from the environment and from the host root, as well as in nitrate transport to NTCs, promoting nodule growth. Under high-nitrate conditions (20 mM), *MtNPF7.6* expression is induced, and an over-accumulation of nitrate due to MtNPF7.6-nitrate-transport inhibits nodule functioning. Interestingly, comparing the transcriptome of wild-type and *mtnpf7.6* nodules, it has been shown that the expression patterns of four genes, encoding MtNRT2.1, MtNRT2.2, MtNRT2.3 and MtNPF6.5, were altered in the mutants, suggesting that MtNPF6.5 and MtNRT2s may be involved in the nutrient or signal exchange in nodule [46]. 

Concerning *P. sativum*, 69 PsNPFs were identified [11]. In addition, a full-length Unigene set of expressed sequences has been developed in *P. sativum* by sequencing 20 cDNA libraries produced from various plant organs harvested at various developmental stages from plants grown under different conditions [13] (https://urgi.versailles.inra.fr/download/pea/Pea_PSCAM_transcriptome, accessed on 15 March 2023). However, some NPFs mentioned in [13] were not identified previously [11]. Thus, to identify the complete PsNPF family in *P. sativum*, we performed a blastp search using PsNPF6.7 (Psat2g025760) as a query against *P. sativum* v1a genomic assembly [10]. We were able to find 90 putative PsNPF sequences (Appendix A), among which we found the 69 previously identified [11] and 21 new ones distributed in the 8 clades (Figure 1) previously described [11]. 

The new sequences are distributed as follows: one sequence belongs to the clade 1, two to the clade 2, one to the clade 3, six to the clade 4, five to the clade 5, two to the clade 7 and four to the clade 8. New PsNPF were annotated according to the two-number code previously established [12]. Then, we exported the expression data of the 90 *PsNPF* genes from the full-length Unigene set of *P. sativum* [13] (Appendix A). It should be noted that the length of PsNPF proteins ranged from 93 to 637 amino acids (Appendix A), with some protein sequences being much shorter than those of NPFs already described in the literature: they have been retained in this study because the corresponding genes are expressed (except *PsNPF5.23*), sometimes very significantly, as seen for *PsNPF4.16* (233 amino acids), which is very strongly expressed in the peduncles of the C stage [13] (Appendix A).

In a study [13], 842 genes were shown to be specifically expressed in nodules. Among them, 66 contigs encoded transporters of various families, of which 6 belonged to the NPF family (Figure 2A), and 3 showed significant expression in nodules but were also expressed in other organs (Figure 2B). One of them, PsNPF7.1, is the ortholog of MtNPF7.6 [46] (Appendix A). *PsNPF7.1* is specifically and very strongly expressed in nodules (Figure 2, [13]). In a recent study, we investigated whether *Rhizobium*-derived signals interfere with nitrate signaling in *P. sativum* [49]. It appeared that *PsNPF7.1* expression was induced in 12-day-old seedlings only in the presence of *Rhizobium*. In addition, *PsNPF7.1* expression was upregulated by 1 mM nitrate and downregulated by 10 mM. A possible role of PsNPF7.1 in nodule functioning dependent on environmental nitrate concentration would be interesting to study further. The orthologous genes of *MtNPF1.7*, *LjNPF8.6* and *LjNPF3.1* in *P. sativum*, *PsNPF1.5*, *PsNPF8.4* and *PsNPF3.1*, respectively, would also be interesting to study (Appendix A). It is worth noting that some *NPF* genes produce different transcripts (Appendix A), as seen for *AtNPF5.5* [50]. Among the genes mentioned above, *PsNPF1.5* produces two different transcripts. It would be interesting to investigate whether the corresponding proteins are both functional and which role they fulfill. 

## 4. NRT2s Playing a Role in Nodule Functioning

LjNRT2.4 was the first NRT2 to be thoroughly studied in *L. japonicus*. In contrast to the other *LjNRT2* genes, a strong induction of *LjNRT2.4* expression was observed in nodules compared to roots [19,20]. A positive role of LjNRT2.4 was reported in a nitrate-mediated nodule functioning pathway [20]. In fact, two *Ljnrt2.4* mutants were impaired in nitrate content and nitrogenase activity in nodules. LjNRT2.4, whose tissue localization was shown to be the nodule vascular bundles and subcellular localization the plasma membrane, would transport nitrate into the N_2_-fixing cells of the nodule. Nitrite derived from nitrate reduction in the cytoplasm can be transported to the mitochondria where it serves as an electron acceptor in the respiratory chain, thus contributing to ATP synthesis [9,51,52]. Nitrate can also be reduced to nitrite by nitrate reductase in the bacteroid. LjNPF8.6, localized in the peribacteroidal membrane, would play a role in the regulation of nitrate flux between the plant cell and the bacteroid [38]. Thus, the model proposed in nodule functioning involves LjNRT2.4 and LjNPF8.6 in a complementary manner [20]. 

LjNRT2.1 has also been studied in depth. Using *Ljnrt2.1* mutants, it has been shown how LjNRT2.1 control root nodule symbiosis in a nitrate-rich environment in *L. japonicus* [21]. The authors proposed a model in which LjNRT2.1 acts in the same signaling pathway as LjNLP1 and LjNLP4 for the nitrate-induced control of nodulation. In the presence of nitrate, the LjNLP1 transcription factor induced *LjNRT2.1* expression. LjNRT2.1 transports nitrate from the soil to the root. The increase of nitrate in the root triggers the nuclear localization of LjNLP4, which inhibits nodulation through the regulation of gene expression. As *LjNLP1* is activated by nitrate, it has been suggested that another nitrate transporter than LjNRT2.1 should be involved in the model to allow the first step, which is nitrate transport and *LjNLP1* activation [21]. In addition, LjNIN, a positive regulator of nodulation, whose expression is induced by rhizobial infection [53,54], would negatively regulate the expression of *LjNRT2.1* resulting in a reduction of nitrate uptake. Thus, LjNRT2.1 would be at the center of a strategy used by the plant regarding nitrate acquisition, switching from dependence on soil nitrate to symbiotic fixation [21]. 

Among the three MtNRT2 of *M. truncatula*, only the role of MtNRT2.1 in nodulation has been addressed [55]. Some similarities between MtNRT2.1 and LjNRT2.1 have been observed. In fact, *MtNRT2.1* expression, like that of *LjNRT2.1*, is activated by MtNLP1. Using *Mtnrt2.1* mutants, it has been shown that MtNRT2.1 encodes a high-affinity nitrate transporter responsible for the majority of nitrate taken up by the plant in the 0.5–5 mM nitrate concentration range [55]. In addition, MtNRT2.1’s ability to uptake nitrate in *Xenopus laevis* oocytes requires MtNAR2. MtNRT2.1 was also shown to play a dual role in nitrate regulation of nodulation in *M. truncatula* as it is required for nodule establishment under low-nitrate conditions and necessary for repression of nodulation under high-nitrate conditions [55]. Accordingly, a model has been proposed in which low nitrate induces *MtCEP1* expression, which systemically induces *MtNRT2.1* expression through MtCRA2, resulting in an enhancement in nodulation and nitrate uptake. MtNLP1, whose localization in the nucleus is limited under low nitrate, is increased by high nitrate in the nucleus, leading to the activation of the expression of *CLE5*, which negatively regulates nodulation [55]. Thus, MtNRT2 has been shown to play a role in nodule functioning in *M. truncatula* as well as MtNAR2 which seems necessary for nitrate transport [55]. The importance of MtNAR2 in nodules seems to be confirmed by its expression in this organ [18].

In the pea genome, only one full-length *PsNRT2*, named *PsNRT2.3* (Ps4g113000), was identified [11]. Two more *PsNRT2* genes exist, *PsNRT2.1* (Psat4g155600) and *PsNRT2.2* (Psat7g149120), but both corresponding proteins are short with only three transmembrane domains against eight in NRT2 in general. A possible loss of nitrate transport function has been suggested for these two proteins [11]. We have made a phylogenetic tree to establish PsNRT2 relationship with NRT2 of *M. truncatula* and *L. japonicus* (Figure 3). It shows a clustering of PsNRT2.1/2.2 with MtNRT2.1/2.2 and LjNRT2.1/2.2 on the one hand, and a clustering of PsNRT2.3 with MtNRT2.3 and LjNRT2.3 on the other hand. We confirm that LjNRT2.4 appears isolated in the phylogenetic tree, having no ortholog in *M. truncatula* [20] and having no ortholog in *P. sativum* either (Figure 3).

The omics data [13] allow visualization of the expression of the three *PsNRT2* genes and of the *PsNAR2* (Psat4g061680) gene under different conditions in different tissues (Figure 4). 

It can be noted that despite the smaller size of PsNRT2.1 and PsNRT2.2 proteins, corresponding genes were expressed (Figure 4A). *PsNRT2.1*, *PsNRT2.2* and *PsNAR2* were very strongly expressed in the roots at two stages (7–8 nodes, 5–6 opened leaves and at beginning of flowering), while *PsNRT2.3* was much less expressed at those stages. The results indicate that *PsNRT2.1* and *PsNRT2.2* were also expressed in nodules but much less than in roots (at least 18-times less), and *PsNRT2.3* was almost not expressed in nodules. The question arises as to which of these PsNRT2 would play a role in nodule functioning in *P. sativum*. In *P. sativum*, no *NRT2* gene is so strongly expressed in nodules as *LjNRT2.4* in *L. japonicus* [19], and there is no ortholog of *LjNRT2.4* in *P. sativum* (Figure 3). Among the three *PsNRT2* genes in pea, *PsNRT2.1* was the most highly expressed in nodules (Figure 4A). As we have seen, *PsNRT2.1* orthologs in *M. truncatula* and *L. japonicus* play an important role in nodule functioning. The spatial expression pattern of *LjNRT2.1* was precisely studied during nodule development using *pLjNRT2.1*:GUS reporter analysis [21]. At the initial developmental stages of nodulation, *LjNRT2.1* was expressed within cortical cells, while at later stages, *LjNRT2.1* was expressed in the outer regions of nodules, including the epidermis. The spatial expression pattern of *MtNR2.1* during nodulation was also determined using *pMtNRT2.1:GUS* transformed *M. truncatula* hairy roots. *MtNRT2.1* is expressed in root vascular tissues and nodule meristem. Further study would be necessary to study *PsNRT2.1* and *PsNRT2.2* expression in detail during nodule development and to see if either or both proteins, PsNRT2.1 and PsNRT2.2, have a role in the regulation of nodulation despite the protein’s smaller size compared with other NRT2s.

## 5. Future Prospects

Identification of the roles of the myriad of putative nitrate transporters expressed in nodules would open new avenues for better characterizing the involvement of nitrate and other substrates, such as phytohormones, transported by members of NPF and NRT2 families in nodules functioning. In fact, besides the well-illustrated role of nitrate as a negative regulator of nodulation through local and systemic signaling pathways [9], nitrate plays an important role as a source of nitric oxide (NO). Interestingly, both the plant and the symbiont were shown to use nitrate as a substrate for NO synthesis in functional nodules [51]. NO has been shown to be produced from early phases of plant–symbiont interaction to nodule senescence [56]. At early phases, NO contributes to the repression of plant defense reactions, which favors the microbe penetration in plant tissue, while in mature nodules, NO participates to the modulation of nitrogen acquisition by inhibiting N_2_ fixation. Nitrate, as a provider of NO, also contributes to the energy status (ATP synthesis) in both nodules and bacteroids through the mitochondrial NO_3_^−^-NO respiration in invaded cells and the denitrification pathway in bacteroids [51]. 

It is thus of importance that the putative transporters of nitrate expressed in nodules be functionally characterized because their contribution seems essential to ensure nitrate trafficking between the root system and nodules and between invaded cells and bacteroids enclosed in symbiosomes [9,51,56]. In this context, an integrative model could be drawn in *L. japonicus*, where complementary roles are proposed for two nitrate transporters: a high-affinity transporter LjNRT2.4 to ensure nitrate allocation to the N_2_-fixing cells [20], and a low-affinity transporter LjNPF8.6 that regulates nitrate flux between plant cell cytosol and bacteroid compartments [38]. Furthermore, NPFs transport other substrates than nitrate, such as phytohormones [57,58], that might be involved in nodule functioning. Auxin and ABA were found to play major roles in nodule formation [42], and GA was reported as a positive regulator of nodule functioning [59]. Thus, NPF transporters could couple nitrate and hormone signaling during root symbiosis. 

## 6. Methods for Generating Expression Data

Data related to gene expression used in the present study derive from supplemental data of the article published by Alves-Carvalho et al. [13] and made available on *The Plant Journal* website. In this chapter, we briefly supply the methods used by authors for generating these data.

### 6.1. Plant Material

The plant material subjected to RNAseq analyses was produced in three independent experiments, with three biological repetitions (for more details, see Table S9 in Reference [13]). Authors used pea cultivar ‘Cameor’, and inoculation, when applicable, was carried out with the P221 *Rhizobium leguminosarum* strain. 

In two experiments, pea plants were inoculated with *R. leguminosarum* and grown in one experiment (named ‘first experiment’ by authors) under hydroponic conditions in glasshouses under either high nitrogen (14 mM) or low nitrogen (0.625 mM) conditions, and in the other experiment (named ‘third experiment’ by authors), in an aeroponic system supplemented with 0.5 mM ammonium nitrate. In the second experiment, seeds were germinated in Petri dishes at 25 °C for 5 days before being transferred to 7 L pots filled with a mix of attapulgite and clay beads and irrigated with a nutrient solution at 14 mM nitrogen.

### 6.2. cDNA Library Preparation and Sequencing

Total RNA was isolated with the RNeasy plant mini kit (Qiagen, Hilden, Germany), and its quality was assessed through a bioanalyzer (Agilent, Santa Clara, CA, USA). Subsequently, poly(A) mRNA was purified using the Dynabeads mRNA purification kit (Thermo Fisher Scientific, Waltham, MA, USA). The synthesis of double-stranded cDNA involved utilizing 0.5–1 μg cDNA per sample for cDNA library preparation. Following fragmentation, the NEBNext DNA Library Prep Master Mix Set for Illumina from New England Biolabs (Ipswich, MA, USA) was employed for library preparation. Illumina’s recommendations were followed for paired-end sequencing (San Diego, CA, USA), with one lane per library. Fourteen libraries were sequenced on the Genome Analyzer II platform, and six libraries were sequenced on the HiSeq2000 platform.

### 6.3. Differential Gene Expression Analyses

Principal component analysis of the RPKMnorm expression values (number of reads per kb per million reads divided by geometric mean of the RPKM for three control genes: histone H1, actin and EF1α) was conducted to compare the transcriptomes of the 20 libraries represented by PsCam_LowCopy and PsCam_HighCopy. Following this, a more detailed analysis of the PsCam_LowCopy gene set was carried out: a heatmap was generated through mean linkage hierarchical clustering of pairwise Pearson correlation coefficients of RPKMnorm in the 20 libraries, utilizing the MULTIEXPERIMENT VIEWER software (https://webmev.tm4.org, accessed on 8 January 2024). To analyze differential gene expression among libraries, K-means hierarchical clustering was performed using Genesis (K = 20, http://genome.tugraz.at/, accessed on 8 January 2024) and DEseq (DESeq Bioconductor package in R). Count data normalization and the identification of differentially expressed contigs were accomplished through pairwise comparisons, employing a negative binomial distribution. The DeSeq analyses included replications in pairwise comparisons of root libraries (RootSys_A_HN, RootSys_A_LN, Root_B_LN and Root_F_LN), nodule libraries (Nodule_G_LN, Nodule_B_LN and Nodule_A_LN) and shoot libraries (Shoot_A_HN, Shoot_A_LN, Leaf_B_LN, LowerLeaf_C_LN and UpperLeaf_C_LN). A false discovery rate threshold of 0.05 (Benjamini and Hochberg method) was applied to identify significantly differentially expressed contigs between pea tissues. The visualization of differentially expressed sequences was performed using TOPGO (http://topgo.bioinf.mpi-inf.mpg.de/, accessed on 8 January 2024).

## 7. Conclusions

A large number of putative NPFs transporters have been identified as expressed in nodules [15,36,60]; however, only few have been thoroughly studied, i.e., two in *M. truncatula*, MtNPF1.7 [42] and MtNPF7.6 [46] and two in *L. japonicus*, LjNPF8.6 [38] and LjNPF3.1 [39]. In *P. sativum*, *PsNPFs* identified in this study as specifically expressed in nodules (Figure 2A) or expressed in nodules and other organs (Figure 2B) are interesting candidates waiting for the functional characterization and investigation of their roles in nodules. Regarding NRT2s, three have been shown to play a role in nodule functioning in *L. japonicus*, LjNRT2.4 [20], LjNRT2.1 [21] and *M. truncatula* MtNRT2.1 [55]. In *P. sativum* genome, three *NRT2* genes were identified: one full-length named *PsNRT2.3* (Ps4g113000), as well as *PsNRT2.1* (*Psat4g155600.1*) and *PsNRT2.2* (*Psat7g149120.1*), both encoding short proteins with only three transmembrane domains against eight in NRT2 in general [11]. Further studies are thus necessary for the functional characterization of these putative high-affinity transporters and their potential involvement in the regulation of nodulation.

## Figures and Tables

**Figure 1 plants-13-00322-f001:**
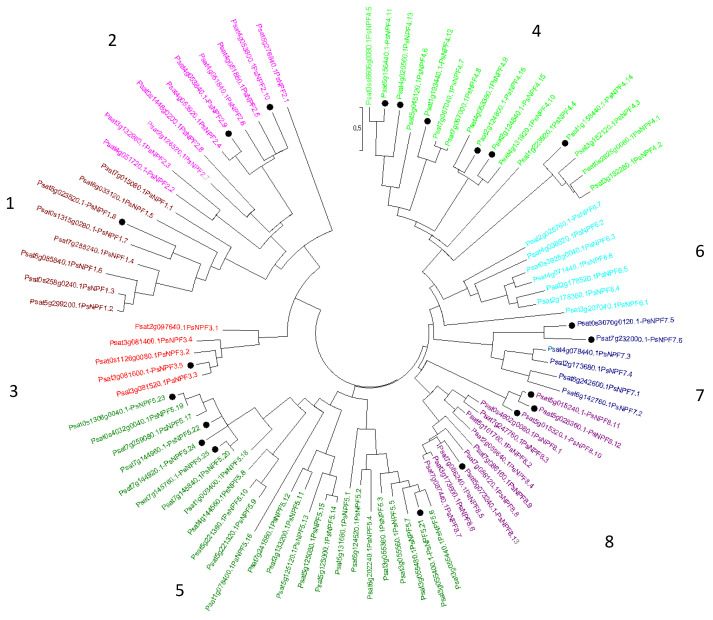
Phylogenetic tree of the NPF family from *P. sativum*. Ninety amino acid sequences were aligned with the CLUSTALW program. The evolutionary history was inferred using the Maximum Likelihood method based on the JTT matrix-based model [47]. Evolutionary analyses were conducted in MEGA7 [48]. The eight NPF clades, numbered from 1 to 8, are indicated by different colors [12]. Tree branches are colored consistently with Appendix A. The newly identified sequences are presented with a black point.

**Figure 2 plants-13-00322-f002:**
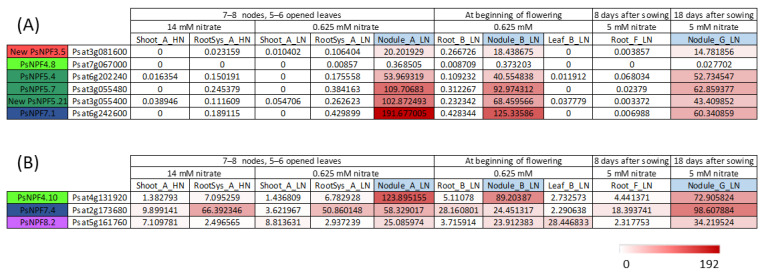
*PsNPF* genes that are expressed in nodules. Data were extracted from the full-length Unigene set of expressed sequences from *P. sativum* [13]. The expression of the 90 *PsNPFs* under all the conditions is presented in the Appendix A. (**A**) *PsNPF* exclusively expressed in nodules. (**B**) *PsNPF* genes highly expressed in nodules and other organs. Numbers are normalized count data.

**Figure 3 plants-13-00322-f003:**
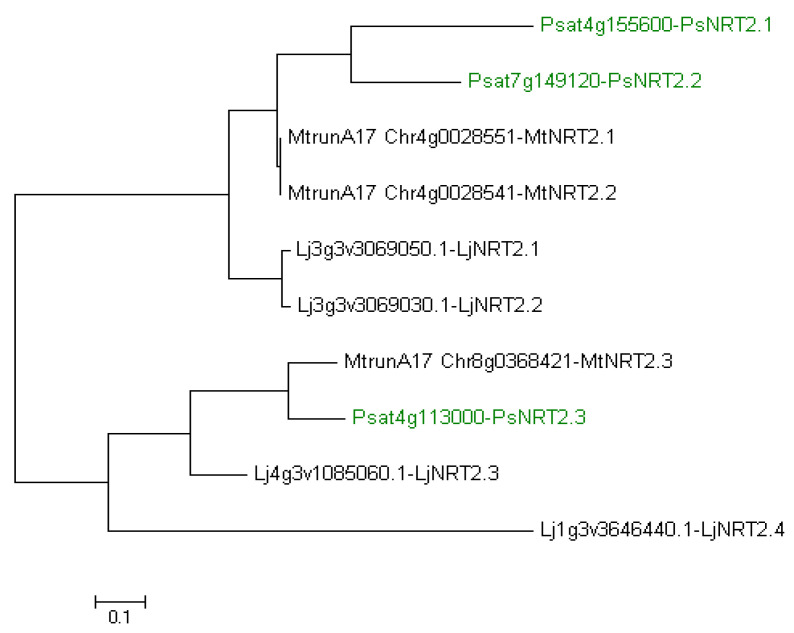
Phylogenetic tree of NRT2 from *Lotus japonicus*, *Medicago truncatula* and *Pisum sativum*. Ten amino acid sequences were aligned with the CLUSTALW program. The evolutionary history was inferred using the Maximum Likelihood method based on the JTT matrix-based model [47]. Evolutionary analyses were conducted in MEGA7 [48]. NRT2 from *P. sativum* are indicated in green. Lj, *L. japonicus*; Mt, *M. truncatula*; Ps, *P. sativum*.

**Figure 4 plants-13-00322-f004:**
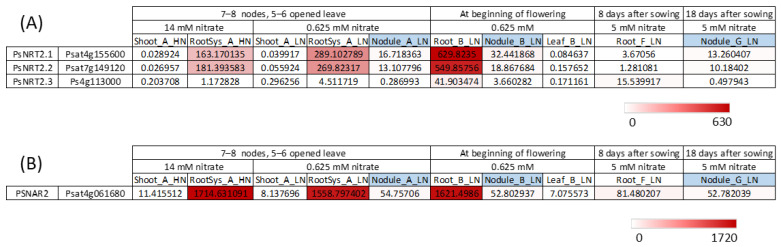
*PsNRT2* and *PsNAR2* genes expression. Data were extracted from the full-length Unigene set of expressed sequences from *P. sativum* resulting from de novo assembly of RNA-seq data [13]. (**A**) *PsNRT2* genes expression. (**B**) *PsNar2* gene expression. Numbers are normalized count data.

## Data Availability

New data created in this study are available in Appendix A.

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
