# Peer review of "NPF and NRT2 from Pisum sativum Potentially Involved in Nodule Functioning: Lessons from Medicago truncatula and Lotus japonicus"

_plants, 2024, doi:10.3390/plants13020322_

Round 1
Reviewer 1 Report
Comments and Suggestions for Authors
I have suggested some changes in the attached PDF, which are self-explanatory.
Abstract: Add a few more concluding sentences at the end of the Abstract.
Introduction: A few sentences should be rephrased.
Please check the Table 1-2 headnote. It is very confusing for the reader. Pl. Modify it.
Table 1-2 missing some data. Pl. check.
In many sentences, you have used a semicolon (;) between the clauses. Please use that/which or similar words to maintain the flow of a sentence.
Conclusion: It is lengthy. Please reduce its size and mention only the salient features related to the study matter.
Conclusion: My suggestion is to avoid references in the conclusion. However, it is up to the editor and other Reviewers how they would take it.

Minor editing of English language required.
Author Response
Answers to Reviewer 1
I have suggested some changes in the attached PDF, which are self-explanatory.
Authors answer: We wish to thank you for your time and valuable suggestions that we took into account for our manuscript revision. Modifications are highlighted in the latest version of the manuscript (please see the attachment).
Abstract: Add a few more concluding sentences at the end of the Abstract.
Authors answer: The abstract was enriched by the addition of interesting detailed information on the transporters of either NPF or NRT2 family that have been well characterized in planta as having effective role in the control of nodules development and/or functioning.
Introduction: A few sentences should be rephrased.
Authors answer: This has been done.
Please check the Table 1-2 headnote. It is very confusing for the reader. Pl. Modify it.
Authors answer: Headnote of Table 1 was changed as requested.
Table 1-2 missing some data. Pl. check.
Authors answer: Thank you for the notice, Table 1 was checked and modified.
In many sentences, you have used a semicolon (;) between the clauses. Please use that/which or similar words to maintain the flow of a sentence.
Authors answer: We agree with the reviewer that in certain instances the use of semicolon may be inappropriate. We checked these cases in the manuscript and modified them to adopt the most appropriate punctuation.
Conclusion: It is lengthy. Please reduce its size and mention only the salient features related to the study matter.
Authors answer: The conclusion was a bit long because it contained some perspectives and openings for future research. To meet the reviewer's request and not “lose” suggestions for future research, we have divided the “Conclusions” chapter into two chapters, “Future Prospects” and a short “Conclusion”.
Conclusion: My suggestion is to avoid references in the conclusion. However, it is up to the editor and other Reviewers how they would take it.
Authors answer: We are sorry, we don’t agree with the reviewer’s opinion. In order to avoid misuse of the literature, we prefer to cite our sources in all instances where we refer to them.
Additionally, for the present article, neither of the other two reviewers asked to avoid citing the literature in the conclusion.

Reviewer 2 Report
Comments and Suggestions for Authors
Abstract
1. The value of this study should be highlighted in the end of abstract;
Introduction
1. NAR2/NRT3 is very important for the NRT2-mediated high-affinity nitrate transport. Please confirm its role in nodulation;
2. The total profile of Table 1 colud not be seen, please improve it;
3. The resolution and word sizes of figures should be enhanced and enlarged;
4. In Arabidopsis, seven NRT2 homologs (NRT2.1-NRT2.7) were identified; please clarify how many members of the whole NRT2 family were identified in this study. Whether all the seven NRT2s are identified in this study, if not, please disccuss it.
5. The language usage is not very good, please polish it by more native English-speakers.
Comments on the Quality of English LanguageThe language had better be polished by native English-speakers to improve its readabilty.
Author Response
Answers to Reviewer 2
Thank you very much for taking the time to review this manuscript. Please find the detailed responses below and the corresponding corrections highlighted in the re-submitted file (please see attachment). Please find a point-by-point response to comments and suggestions.
Abstract
- The value of this study should be highlighted in the end of abstract;
Authors answer: The abstract was enriched by the addition of interesting details on the transporters of either NPF or NRT2 family that have been well characterized in planta as having effective role in the control of nodules development and/or functioning. Also, the last sentences value the originality of the present review.
Introduction
- NAR2/NRT3 is very important for the NRT2-mediated high-affinity nitrate transport. Please confirm its role in nodulation;
Authors answer: Thank you for this interesting remark, we agree that it is important to refer to the role of NAR2/NRT3 in nodulation because it is determinant for the function of NRT2 as a transporter. Thus according the reviewer’s request, in page 6 line 264-265 and from lines 273 to 275, we refer to original works on NAR2/NRT3 in legumes and in relation to nodulation.
- The total profile of Table 1 could not be seen, please improve it;
Authors answer : This has been done.
- The resolution and word sizes of figures should be enhanced and enlarged;
Authors answer: The request was taken into account, the figures have been modified and their resolution improved.
- In Arabidopsis, seven NRT2 homologs (NRT2.1-NRT2.7) were identified; please clarify how many members of the whole NRT2 family were identified in this study. Whether all the seven NRT2s are identified in this study, if not, please discuss it.
Authors answer: In the present study all NRT2s expressed in the model legumes, M. truncatula, L. japonicus and P. sativum that are either functionally characterized as involved in nodules or at least proposed as putative candidates to play a role in nodules are identified and discussed.
- The language usage is not very good, please polish it by more native English-speakers.
Authors answer: This has been done.
Reviewer 3 Report
Comments and Suggestions for Authors
This is a very interesting and informative review. The aims and objectives were well laid down in the introduction lines 47-56. Also, the conclusions were well aligned with those objectives ascertaining future lines of research about this biological nitrogen acquisition and its environment.
Author Response
Answers to Reviewer 3
This is a very interesting and informative review. The aims and objectives were well laid down in the introduction lines 47-56. Also, the conclusions were well aligned with those objectives ascertaining future lines of research about this biological nitrogen acquisition and its environment.
Authors answer: We warmly thank the reviewer for taking the time to review this manuscript and for its encouraging opinion on our work. Please see the attachment.
